# Improving LLM General Preference Alignment via Optimistic Online Mirror Descent

**Yuheng Zhang** *
UIUC

**Dian Yu**
Tencent AI Lab

**Tao Ge**
Tencent AI Lab

**Linfeng Song**
Tencent AI Lab

**Zhichen Zeng**
UIUC

**Haitao Mi**
Tencent AI Lab

**Nan Jiang**
UIUC

**Dong Yu**
Tencent AI Lab

## Abstract

Reinforcement learning from human feedback (RLHF) has demonstrated remarkable effectiveness in aligning large language models (LLMs) with human preferences. Many existing alignment approaches rely on the Bradley-Terry (BT) model assumption, which assumes the existence of a ground-truth reward for each prompt-response pair. However, this assumption can be overly restrictive when modeling complex human preferences. In this paper, we drop the BT model assumption and study LLM alignment under general preferences, formulated as a two-player game. Drawing on theoretical insights from learning in games, we integrate optimistic online mirror descent into our alignment framework to approximate the Nash policy. Theoretically, we demonstrate that our approach achieves an $\mathcal{O}(T^{-1})$ bound on the duality gap, improving upon the previous $\mathcal{O}(T^{-1/2})$ result. Meanwhile, it enjoys a linear convergence rate in the last iterate, a property not achieved by previous methods. More importantly, we implement our method and show through experiments that it outperforms state-of-the-art RLHF algorithms across multiple representative benchmarks.

## 1 Introduction and Related Works

Reinforcement learning from human feedback (RLHF) has played a pivotal role in aligning large language models (LLMs) with human preferences. The goal of RLHF is to fine-tune LLMs to generate responses that are preferred by humans. It has been successfully deployed in state-of-the-art models, including Instruct-GPT [Ouyang et al., 2022] and Claude [Bai et al., 2022b]. The first RLHF framework for LLMs was developed by Ouyang et al. [2022], where after the pre-training stage, the LLM is fine-tuned to maximize the reward signal from a reward model using the proximal policy optimization (PPO) algorithm [Schulman et al., 2017]. This pipeline requires training both the reward model and the policy model. In addition, policy gradient approaches such as PPO often exhibit high variance and instability during training [Peng et al., 2023], leading to increased computational costs.

To develop a more stable and computationally lightweight alignment approach, Rafailov et al. [2024b] propose the Direct Preference Optimization (DPO) algorithm, which directly trains the LLM on a preference dataset and bypasses the need for a reward model. DPO uses an offline preference dataset, and since its development, a line of research has explored different exploration strategies and proposed online direct preference alignment algorithms [Xiong et al., 2024, Xie et al., 2024, Dong et al., 2024, Yuan et al., 2024]. All these methods assume that human preferences can be modeled using the Bradley-Terry (BT) model, where a reward function $R^*$ exists such that, for any prompt $x$

---

*Email: yuhengz2@illinois.edu.

39th Conference on Neural Information Processing Systems (NeurIPS 2025).

and response pair $(y^1, y^2)$, the preference between $y^1$ and $y^2$ satisfies:

$$\mathbb{P}(y^1 \succ y^2 \mid x) = \sigma(R^*(x, y^1) - R^*(x, y^2)),$$

where $\sigma(z) = \frac{1}{1+\exp(-z)}$ is the sigmoid function.

However, the existence of a reward function and the BT model are strong assumptions that can be overly restrictive when modeling complex human preferences. For example, the preference signals in the BT model are always transitive: if $A$ is preferred to $B$ and $B$ is preferred to $C$, then $A$ must always be preferred to $C$. This transitive property contradicts evidence from human decision-making [May, 1954, Tversky, 1969], especially when preferences are at the population level and aggregated from different human groups [May, 1954, Ye et al., 2024]. Furthermore, the limitations of the BT model have also been observed in RLHF practice. Jiang et al. [2023] show that a preference model with 0.4B parameters achieves performance comparable to Llama-2-13B-based reward models. Ye et al. [2024] train a BT reward model and a preference model separately using the same base model and preference dataset, and their results demonstrate that the preference model consistently outperforms the reward model on Reward-Bench [Lambert et al., 2024] under different base models. These findings motivate us to drop the BT model assumption and instead consider general preferences.

In this work, we study the problem of aligning LLMs with general preferences and formulate it as a two-player zero-sum game. Our objective is to approximate the Nash policy of the game, which ensures a win rate of at least 50% against any other policy. As established in the game theory literature [Bai et al., 2020, Liu et al., 2021], self-play algorithms have proven to be highly effective in approximating Nash policies. Building on this, we aim to propose a novel online RLHF algorithm that further leverages the self-play strcture to enhance general preference alignment for LLMs. Our contributions are summarized as follows.

**Contributions.** We propose a novel online general preference alignment algorithm, Optimistic Nash Policy Optimization (ONPO). Inspired by recent advancements in game theory, our algorithm integrates optimistic online mirror descent [Rakhlin and Sridharan, 2013, Syrgkanis et al., 2015] into the self-play framework. By utilizing a reward predictor in a two-step update strategy, ONPO more effectively leverages the self-play mechanism and achieves a faster convergence rate of $\mathcal{O}(T^{-1})$ on the duality gap, improving upon the previous $\mathcal{O}(T^{-1/2})$ result. Moreover, ONPO enjoys a linear convergence rate in the last iterate, a property not achieved by previous methods such as INPO [Zhang et al., 2024].

ONPO can be efficiently implemented by directly minimizing a loss objective on a preference dataset, making it computationally lightweight in practice. We evaluate ONPO on several representative benchmarks, comparing it with state-of-the-art general preference alignment algorithms. Experimental results demonstrate that ONPO consistently outperforms or achieves performance comparable to the baselines across different base models and benchmarks. Notably, on the AlpacaEval 2.0 benchmark [Li et al., 2023a], ONPO achieves a 21.2% and 9.9% relative improvement over the strongest baseline when using Mistral-Instruct and Llama-3-8B as the base models, respectively.

## 2 Preliminary

**Problem Setup.** We study the contextual formulation which is extensively used in previous RLHF literature [Rafailov et al., 2024b, Xiong et al., 2024]. The prompt $x \in \mathcal{X}$ is sampled from an unknown prompt distribution $d_1$. $\mathcal{Y}$ is the response space and an LLM is characterized by a policy $\pi : \mathcal{X} \to \Delta(\mathcal{Y})$ which outputs the response probability given the context. For any policy $\pi$, we use $\mathbb{E}_\pi$ to denote the expectations under $\pi$.

**General Preferences.** In this work, we drop the BT model assumption [Bradley and Terry, 1952] and focus on directly aligning LLMs with general preferences. To this end, we define a general preference oracle as follows:

**Definition 1** (General Preference Oracle). There exists a preference oracle $\mathbb{P} : \mathcal{X} \times \mathcal{Y} \to \mathcal{Y} \to [0, 1]$, which can be queried to obtain the binary preference signal:

$$z \sim \text{Ber}\big(\mathbb{P}(y^1 \succ y^2 \mid x)\big),$$

where $z = 1$ indicates $y^1$ is preferred to $y^2$, and $z = 0$ indicates the opposite.

Unlike the BT model assumption, which assumes the existence of a reward function $R^*$ for each $x$ and $y$, the general preference oracle always compares $y^1$ to another $y^2$. This setup aligns with practical scenarios, where it is often easier for users to compare two responses than to assign an absolute score to a single response. Since the preference signal always involves two responses, potentially come from two different policies, we formulate the LLM alignment problem as a two-player zero-sum game. The objective of this game is the expected win rate between the two players:

$$J(\pi_1, \pi_2) := \mathbb{E}_{x \sim d_1} \mathbb{E}_{y^1 \sim \pi_1, y^2 \sim \pi_2} \left[ \mathbb{P}(y^1 \succ y^2 \mid x) \right].$$

Here $\pi_1$ is the policy of the max-player, aiming to maximize the objective, while $\pi_2$ is the policy of the min-player, aiming to minimize it.

**Nash Policies and Duality Gap.** Our learning goal is to find the Nash equilibrium of the game, which is defined as:

$$\pi_1^*, \pi_2^* := \operatorname*{argmax}_{\pi_1} \operatorname*{argmin}_{\pi_2} J(\pi_1, \pi_2).$$

Due to the symmetric nature of the game, the Nash policies for both players are identical, i.e., $\pi_1^* = \pi_2^* = \pi^*$, and the game value is $J(\pi^*, \pi^*) = 0.5$. Since Nash policies are the best responses to each other, for any policy $\pi$, we have $J(\pi^*, \pi) \geq 0.5$, indicating that the Nash policy will not lose to any other policy. To quantify how well a policy $\pi$ approximates $\pi^*$, we define the duality gap as:

$$\mathrm{DualGap}(\pi) := \max_{\pi_1} J(\pi_1, \pi) - \min_{\pi_2} J(\pi, \pi_2).$$

The duality gap is non-negative and $\mathrm{DualGap}(\pi) = 0$ if and only if $\pi = \pi^*$. Hence, our goal is to find a policy that minimizes the duality gap. Once we achieve $\mathrm{DualGap}(\pi) \leq \epsilon$, we say that $\pi$ is an $\epsilon$-approximate Nash policy.

## 3 Algorithm

In this section, we begin by briefly reviewing the self-play algorithm with online mirror descent (OMD) updates, which is used in previous general preference alignment algorithm [Zhang et al., 2024]. Next, we present our proposed algorithm, which leverages the faster convergence properties of optimistic OMD, inspired by advancements in game theory [Rakhlin and Sridharan, 2013, Syrgkanis et al., 2015]. Through theoretical analysis, we show that our approach achieves an improved bound on the duality gap and a linear convergence rate in the last iterate. Finally, we describe the implementation of our algorithm. Following Azar et al. [2024], Zhang et al. [2024], we omit the context $x$ throughout the rest of the paper since each context is independent.

### 3.1 Self-play Algorithm with OMD Update

Self-play algorithms are widely used in approximating the Nash policy [Bai et al., 2020, Liu et al., 2021]. The key idea is to let the policy play against itself, enabling iterative self-improvement. The algorithm is performed in an online manner, with each iteration using online mirror descent (OMD) to update the policy. Specifically, at iteration $t$, we find the policy that maximizes the following objective:

$$\pi_{t+1} = \operatorname*{argmax}_{\pi} \langle \pi, r_t \rangle - \frac{1}{\eta} \mathrm{KL}(\pi \| \pi_t), \tag{1}$$

where $r_t(y) = \mathbb{P}(y \succ \pi_t) = \mathbb{E}_{y' \sim \pi_t}[\mathbb{P}(y \succ y')]$ is the expected win rate of response $y$ against the current policy $\pi_t$, and $\eta > 0$ is the learning rate. This objective ensures that $\pi_{t+1}$ not only aims to maximize the win rate over $\pi_t$ but also remains close to $\pi_t$, as measured by the KL divergence term. The stability introduced by the KL regularization is critical for achieving a sublinear regret bound. Without this regularization, one can construct examples where the algorithm suffers from linear regret, which is undesirable [Lattimore and Szepesvári, 2020].

We can show that the uniform mixture of $\pi_{1:T}$ achieves an $\mathcal{O}(T^{-1/2})$ duality gap, as stated in the following theorem. The proof is deferred to Appendix C.1.

**Theorem 1.** *Let $D = \max_\pi \text{KL}(\pi \| \pi_1)$ and $\bar{\pi} = \text{unif}\{\pi_1, \cdots, \pi_T\}$. Self-play algorithm in Eq.* (1) *with $\eta = \sqrt{\frac{D}{T}}$ satisfies:*

$$\text{DualGap}(\bar{\pi}) \leq \frac{4\sqrt{D}}{\sqrt{T}}.$$

In RLHF practice, we typically use a small number of iterations (e.g., $T = 3$), so the uniform mixture policy $\bar{\pi}$ can be directly deployed. Unlike Zhang et al. [2024], which adopts a KL-regularized game formulation, we directly use the win rate between two policies as the game objective. This formulation has two advantages: 1. The Nash policy in our formulation guarantees at least a 50% win rate against any other policy, which aligns directly with the goal of general preference alignment. In contrast, the Nash policy in the KL-regularized game only ensures this when the KL terms are negligible. 2. In the KL-regularized setting, the analysis of the OMD algorithm relies on a coverage assumption that the log-density ratio $\log \pi(y) / \log \pi_{\text{ref}}(y)$ is uniformly bounded for all $\pi$, and the regret bound depends linearly on the coverage coefficient. Our analysis avoids this assumption and its associated dependence by directly optimizing the win rate, resulting in improved results.

### 3.2 Optimistic Nash Policy Optimization

While self-play with OMD update already achieves an $\mathcal{O}(\sqrt{T})$ regret bound, which is near-optimal in many online learning scenarios, there is still room for improvement by better leveraging the self-play structure. Recent advancements in learning in games [Rakhlin and Sridharan, 2013, Syrgkanis et al., 2015] demonstrate that a faster convergence rate of $\mathcal{O}(T^{-1})$ can be achieved when both players adopt optimistic OMD update. In this subsection, we introduce how to integrate optimistic OMD into the self-play algorithm, resulting in an algorithm called Optimistic Nash Policy Optimization (ONPO).

The key idea of optimistic OMD is to incorporate a reward or loss predictor at each iteration. Recall that in OMD update, we use the expected win rate over the current policy $\pi_t$ as the reward vector $r_t$ to compute $\pi_{t+1}$. While in optimistic OMD, the learner utilizes a reward predictor $m_t$ and adopts a two-step update strategy:

$$\pi_t = \operatorname*{argmax}_\pi \langle \pi, m_t \rangle - \frac{1}{\eta} \text{KL}(\pi \| \pi_t')$$

$$\pi_{t+1}' = \operatorname*{argmax}_\pi \langle \pi, r_t \rangle - \frac{1}{\eta} \text{KL}(\pi \| \pi_t').$$

Here $\pi_t$ aims to maximize the reward predictor $m_t$ and the auxiliary policy $\pi_{t+1}'$ is updated after observing the actual reward $r_t$. The word "optimistic" comes from that the learner believes that the predictor $m_t$ provides a good approximation of the true reward $r_t$.

Next, we describe how to apply optimistic OMD in our self-play algorithm. In both OMD and optimistic OMD, the KL regularization term is consistently used to ensure that the next policy remains close to the previous policies. This regularization provides stability, making it reasonable to assume that the change from $\pi_t$ to $\pi_{t+1}$ is small. Based on this observation, we directly use the reward information from the previous iteration as the predictor, i.e., let $m_t = r_{t-1} = \mathbb{E}_{y' \sim \pi_{t-1}} [\mathbb{P}(y \succ y')]$.

In the following theorem, we demonstrate that ONPO achieves an $\mathcal{O}(1/T)$ duality gap, improving over the previous $\mathcal{O}(1/\sqrt{T})$ result.

**Theorem 2.** *Let $D = \max_\pi \text{KL}(\pi \| \pi_1')$ and $\bar{\pi} = \text{unif}\{\pi_1, \cdots, \pi_T\}$, ONPO algorithm with $\eta = \min\{\frac{1}{2}, \sqrt{D}\}$ satisfies:*

$$\text{DualGap}(\bar{\pi}) \leq \frac{4\sqrt{D}}{T}.$$

Here, $\pi_1' = \pi_1$ is the initialization policy. Theoretically, $\pi_1'$ can be set as a uniform policy, in which case $D$ is bounded by $\log |\mathcal{Y}|$. In RLHF practice, $\pi_1'$ is typically a supervised fine-tuned policy.

The proof is provided in Appendix C.2. The key to achieving the $\mathcal{O}(1/T)$ rate lies in the regret bounded by variation in utilities (RVU) property of optimistic OMD. Specifically, the stability terms $\|r_t - r_{t-1}\|_\infty^2$ are canceled out by the negative term $-\|\pi_t - \pi_{t-1}\|_1^2$, which arises from the self-play mechanism where $r_t$ represents the win rate over $\pi_t$. Additionally, the stability inherent in

optimistic OMD ensures that the learned policy remains close to the initial policy. This aligns with the motivation behind incorporating KL regularization into the game objective in prior works [Munos et al., 2023, Zhang et al., 2024]. Since our update rule already implicitly enforces this stability, explicit regularization in the game objective is unnecessary.

Although the uniform mixture policy is implementable, a more common choice in RLHF practice is to directly deploy the last policy. In the following theorem, we show that ONPO also achieves a linear convergence rate in the last iterate.

**Theorem 3.** *Assume that the Nash policy $\pi^*$ is unique, with $\eta \leq \frac{1}{8}$, we have $\mathrm{KL}(\pi^* \| \pi_t) \leq \mathcal{O}(C^{-t})$ where $C > 1$ is a constant.*

The proof follows directly from the analysis of Theorem 3 in Wei et al. [2020]. Zhang et al. [2024] also demonstrate that self-play with OMD achieves last-iterate convergence. However, their result relies on the strong convexity induced by the KL regularization terms in their game objective and does not apply to our formulation. This highlights another key advantage of using optimistic OMD: it not only improves the duality gap bound but also ensures last-iterate convergence without requiring explicit regularization.

### 3.3 Implementation of ONPO

In this subsection, we describe the implementation of ONPO with query access to the preference oracle $\mathbb{P}$. The primary challenge in implementing ONPO lies in computing $r_t(y)$, which involves taking an expectation over the entire policy $\pi_t$. Fortunately, this challenge can be addressed by avoiding the direct estimation of $r_t(y)$ and instead relying on binary preference feedback between responses.

To achieves this, our goal is to design a loss function that does not involve $\mathbb{P}(y \succ \pi_t)$ for policy optimization. We focus on obtaining the loss objective for $\pi_t$ here and the derivation for $\pi'_t$ is similar. The key observation is that, $\pi_t$ has a closed-form solution which satisfies $\forall y, y' \in \mathcal{Y}$,

$$\log \frac{\pi_t(y)}{\pi_t(y')} - \log \frac{\pi'_t(y)}{\pi'_t(y')} = \eta \left( \mathbb{P}(y \succ \pi_{t-1}) - \mathbb{P}(y' \succ \pi_{t-1}) \right).$$

Therefore, similar to the techniques used in Azar et al. [2024], Zhang et al. [2024], solving $\pi_t$ is equivalent to finding the minimizer of the following loss function:

$$\mathbb{E}_{y,y' \sim \pi_{t-1}} \left[ \left( g_t(\pi, y, y') - \eta \left( \mathbb{P}(y \succ \pi_{t-1}) - \mathbb{P}(y' \succ \pi_{t-1}) \right) \right)^2 \right].$$

where $g_t(\pi, y, y') = \log \frac{\pi(y)}{\pi(y')} - \log \frac{\pi'_t(y)}{\pi'_t(y')}$. Since the inside win rate term is with respect to $\pi_{t-1}$ and we also have an expectation over $\pi_{t-1}$ outside, the loss function can be further written as

$$\mathbb{E}_{y,y' \sim \pi_{t-1}, y_w, y_l \sim \lambda_p(y,y')} \left[ \left( g_t(\pi, y_w, y_l) - \frac{\eta}{2} \right)^2 \right],$$

where $\lambda_p$ is the preference distribution [Calandriello et al., 2024]:

$$\lambda_p(y, y') = \begin{cases} (y, y') & \text{with probability } \mathbb{P}(y \succ y') \\ (y', y) & \text{with probability } 1 - \mathbb{P}(y \succ y'). \end{cases}$$

To calculate the loss function, we only need the access to sample from the current policy, which is standard and easy to implement in practice. Putting everything together, the implementation of ONPO is summarized in Algorithm 1.

In the beginning, we initialize $\pi'_1$ and $\pi_1$ with the supervised fine-tuned policy $\pi_{\text{SFT}}$. At each iteration $t$, we sample responses from the current policy $\pi_t$ and use the preference feedback from the oracle $\mathbb{P}$ to construct the dataset $D_t$. Then we can directly minimize the corresponding loss functions on $D_t$ to find $\pi'_{t+1}$ and $\pi_{t+1}$ respectively. We use the last iteration policy $\pi_T$ as the output policy, which is consistent with online RLHF practice [Dong et al., 2024, Wu et al., 2024, Zhang et al., 2024].

## 4 Discussion

In this section, we discuss the differences between ONPO and other general preference alignment methods.

---

**Algorithm 1** Implementation of ONPO

---

1: **Input:** Number of iterations $T$, learning rate $\eta$, preference oracle $\mathbb{P}$, supervised fine-tuned policy $\pi_{\text{SFT}}$.
2: Initialize $\pi'_1 \leftarrow \pi_{\text{SFT}}$, $\pi_1 \leftarrow \pi_{\text{SFT}}$.
3: **for** iteration $t = 1, 2, \ldots, T-1$ **do**
4:    Sample response pairs from the current policy $\pi_t$: $\{y_1^{(i)}, y_2^{(i)}\}_{i=1}^n \sim \pi_t$.
5:    Construct preference dataset $D_t = \{y_w^{(i)}, y_l^{(i)}\}_{i=1}^n$ with feedback from the oracle $\mathbb{P}$.
6:    Calculate $\pi'_{t+1}$ as:

$$\pi'_{t+1} = \operatorname*{argmin}_{\pi} \mathbb{E}_{y_w, y_l \sim D_t} \left[ \left( g_t(\pi, y_w, y_l) - \frac{\eta}{2} \right)^2 \right].$$

7:    Calculate $\pi_{t+1}$ as:

$$\pi_{t+1} = \operatorname*{argmin}_{\pi} \mathbb{E}_{y_w, y_l \sim D_t} \left[ \left( g_{t+1}(\pi, y_w, y_l) - \frac{\eta}{2} \right)^2 \right].$$

8: **end for**
9: Output $\pi_T$.

---

**IPO.** Azar et al. [2024] is the first to address general preference alignment in LLMs. The optimization objective of IPO is:

$$\max_{\pi} \mathbb{E}_{y \sim \pi, y' \sim \mu} \left[ \mathbb{P}(y \succ y') \right] - \tau \mathrm{KL}(\pi \| \pi_{\text{ref}}),$$

where $\mu$ is a fixed policy. From a game-theoretic perspective, the goal of IPO is to find the best response to $\mu$. However, this approach only ensures that the learned policy outperforms $\mu$, which leaves the possibility that another policy could outperform the learned policy. In contrast, our approach focuses on learning the Nash policy in a two-player game. This provides stronger theoretical guarantees, as the Nash policy will not lose to any other policy.

**Nash-MD.** Munos et al. [2023] is the first to formulate the alignment problem as a two-player zero-sum game. Their game objective includes KL regularization terms, which ensure that the player's policy remains close to the reference policy $\pi_{\text{ref}}$. The KL terms are weighted by a parameter $\tau$. They proposed an iterative algorithm, Nash-MD, to learn the Nash policy of the game. At each iteration $t$, the policy is updated as:

$$\pi_{t+1} = \operatorname*{argmax}_{\pi} \mathbb{P}(\pi \succ \pi'_t) - \frac{1}{\eta_t} \mathrm{KL}(\pi, \pi'_t),$$

where $\pi'_t$ is a geometric mixture policy of the current policy $\pi_t$ and the reference policy $\pi_{\text{ref}}$:

$$\pi'_t(y) = \frac{\pi_t(y)^{1 - \eta_t \tau} \pi_{\text{ref}}(y)^{\eta_t \tau}}{\sum_{y'} \pi_t(y')^{1 - \eta_t \tau} \pi_{\text{ref}}(y')^{\eta_t \tau}}.$$

Nash-MD requires sampling from the mixture policy $\pi'_t$. However, the response space $\mathcal{Y}$ is often exponentially large, making the exact computation of $\pi'_t$ intractable. To address this, Munos et al. [2023] propose sampling from an approximate policy. The theoretical guarantees of this approximation remain unclear. In contrast, our approach only requires sampling from the current policy $\pi_t$, which is straightforward to implement in practice.

**Online IPO.** Calandriello et al. [2024] propose the online IPO population loss:

$$\mathbb{E}_{\substack{y, y' \sim \text{SG}[\pi] \\ y_w, y_l \sim \lambda_p(y, y')}} \left[ \left( \log \frac{\pi(y_w)\pi_{\text{ref}}(y_l)}{\pi(y_l)\pi_{\text{ref}}(y_w)} - \frac{1}{2\tau} \right)^2 \right],$$

where SG is the stop-gradient operator, which prevents gradients from propagating through the data-generation process. Unlike the offline IPO approach, which always samples from a fixed policy $\mu$, online IPO leverages responses generated by the current policy $\pi$.

Since the policy $\pi$ is updated throughout training, policy gradient methods are used to minimize the objective. However, as discussed earlier, policy gradient methods in RLHF have limitations, including being resource-intensive and unstable to train. In contrast, ONPO avoids these challenges by directly minimizing a loss function over a preference dataset, offering a more stable and efficient implementation.

**DNO.** The theoretical version of DNO (Algorithm 1 in Rosset et al. [2024]) relies on computing $r_t(y) = \mathbb{E}_{y' \sim \pi_t} [\mathbb{P}(y \succ y')]$, which requires taking an expectation over the current policy $\pi_t$. This computation is challenging to implement in practice, so Rosset et al. [2024] propose a practical version, DNO-Prct (Algorithm 2), where $\pi_{t+1}$ is updated as follows:

$$\underset{\pi}{\arg\max} \, \mathbb{E}_{y_w, y_l \sim D_t} \log \left[ \sigma \left( \eta \log \frac{\pi(y_w)\pi_t(y_l)}{\pi_t(y_w)\pi(y_l)} \right) \right].$$

When constructing the dataset $D_t$, only response pairs with large margins are selected. This selection is motivated by the fact that, to approximate DNO, the ideal condition is $\sigma(r_t(y_w) - r_t(y_l)) \approx 1$. However, this cannot be fully achieved since $r_t(y) \in [0, 1]$. Notably, the objective of DNO-Prct is identical to the DPO objective [Rafailov et al., 2024b]. Therefore, DNO-Prct can be viewed as an iterative version of DPO.

**SPPO.** Wu et al. [2024] propose a self-play algorithm SPPO. The policy update in SPPO is:

$$\pi_{t+1} = \underset{\pi}{\arg\min} \, \mathbb{E}_{y \sim \pi_t} \left( \log \frac{\pi(y)}{\pi_t(y)} - \eta \left( \widehat{P}(y \succ \pi_t) - \frac{1}{2} \right) \right)^2,$$

where $\widehat{P}$ is a heuristic approximation of $\mathbb{P}(y \succ \pi_t)$. However, obtaining an accurate estimation of $\mathbb{P}(y \succ \pi_t)$ is challenging in practice. For example, Hoeffding's inequality suggests that more than 100 queries are needed to ensure $\left| \mathbb{P}(y \succ \pi_t) - \widehat{P}(y \succ \pi_t) \right| \leq 0.1$. This requirement results in high annotation and computation costs, as 100 oracle queries are needed for a single response $y$. In contrast, ONPO bypasses the need to estimate $\mathbb{P}(y \succ \pi_t)$ and instead relies on binary preference signals between two responses.

**INPO.** Zhang et al. [2024] propose a self-play algorithm, INPO, which employs OMD to iteratively update the policy, as described in Section 3.1. Leveraging the faster convergence properties of optimistic OMD, ONPO achieves an improved duality gap bound of $\mathcal{O}(T^{-1})$, compared to the $\mathcal{O}(T^{-1/2})$ bound of INPO.

**Concurrent Works.** Two concurrent works also study general preference alignment. Liu et al. [2024] propose a two-level algorithmic framework that differs significantly from ours. In each iteration, their meta-algorithm defines a new KL-regularized game and applies a general preference alignment algorithm, such as INPO, to learn its Nash policy. Their theoretical analysis shows that the resulting policy converges asymptotically to the true Nash policy. Wu et al. [2025] focus on the multi-turn setting and adopt a deep reinforcement learning approach based on the actor-critic framework. In contrast, our work targets a computationally lightweight algorithm and therefore focuses on the single-turn setting.

## 5 Experiments

### 5.1 Main Results

**Experiment Setup.** We implement ONPO following the online RLHF workflow described in Dong et al. [2024]. Two base models are used as the initial policy $\pi_1$: Llama-3-SFT[2], based on Llama-3-8B [Dubey et al., 2024], and Mistral-Instruct-v0.3[3], an instruct fine-tuned version of the Mistral-7B-v0.3. For the general preference oracle, we use a pairwise preference model[4], which demonstrates better performance compared to the BT reward model [Zhang et al., 2024]. Training details for the preference model are available in Dong et al. [2024].

---

[2] https://huggingface.co/RLHFlow/LLaMA3-SFT

[3] https://huggingface.co/mistralai/Mistral-7B-Instruct-v0.3

[4] https://huggingface.co/RLHFlow/pair-preference-model-LLaMA3-8B

Table 1: Results on three benchmarks. "ONPO+Mistral-It" refers to tuning the Mistral-Instruct model with ONPO, while "ONPO+Llama-3-SFT" refers to tuning the Llama-3-SFT model with ONPO. Results where the baseline outperforms ONPO are underlined.

| Model | Size | AlpacaEval 2.0 | Arena-Hard | MT-Bench |
|---|---|---|---|---|
| Iterative DPO + Mistral-It | 7B | 32.0 | 22.2 | 7.35 |
| SPPO + Mistral-It | 7B | 33.1 | 24.5 | 7.51 |
| INPO + Mistral-It | 7B | 35.3 | 25.3 | 7.46 |
| ONPO + Mistral-It | 7B | **42.8** | **29.7** | **7.68** |
| Iterative DPO + Llama-3-SFT | 8B | 28.3 | 31.9 | 8.34 |
| SPPO + Llama-3-SFT | 8B | 38.5 | 32.9 | 8.23 |
| INPO + Llama-3-SFT | 8B | 44.2 | _37.0_ | 8.28 |
| ONPO + Llama-3-SFT | 8B | **48.6** | **36.4** | **8.40** |
| Llama-3-8B-it | 8B | 24.8 | 21.2 | 7.97 |
| Tulu-2-DPO-70B | 70B | 21.2 | 15.0 | 7.89 |
| Llama-3-70B-it | 70B | 34.4 | 41.1 | 8.95 |
| Mixtral-8x22B-it | 141B | 30.9 | 36.4 | 8.66 |
| GPT-3.5-turbo-0613 | - | 22.7 | 24.8 | 8.39 |
| GPT-4-0613 | - | 30.2 | 37.9 | 9.18 |
| Claude-3-Opus | - | 40.5 | 60.4 | 9.00 |
| GPT-4 Turbo (04/09) | - | 55.0 | 82.6 | - |

At each iteration, the current policy generates $K = 8$ responses using a different set of prompts[5]. To select $y_w$ (winner) and $y_l$ (loser), we follow the tournament approach in Zhang et al. [2024], where the eight responses are compared pairwise to identify the winning and losing responses.

Since online or iterative alignment methods have been shown to outperform offline counterparts, we focus on comparing ONPO with other online methods for a fair evaluation. These include iterative DPO [Dong et al., 2024], SPPO [Wu et al., 2024] and INPO [Zhang et al., 2024], where the latter two are general preference alignment approaches.

We evaluate the models on three representative benchmarks: AlpacaEval 2.0 [Li et al., 2023a], Arena-Hard [Li et al., 2024] and MT-Bench [Zheng et al., 2024]. AlpacaEval 2.0 has 805 instructions from five datasets, including self-instruct test set [Wang et al., 2022], Open Assistant test set, Anthropic's helpful test set [Bai et al., 2022b], Vicuna test set [Zheng et al., 2024] and Koala test set [Geng et al., 2023]. Arena-Hard includes 500 challenging user queries from Chatbot Arena. Both AlpacaEval 2.0 and Arena-Hard compare model-generated answers against reference answers from a baseline model, using GPT-4 Preview-1106 as the judge model. We report the win rate for Arena-Hard and the length-controlled (LC) win rate [Dubois et al., 2024] for AlpacaEval 2.0. MT-Bench consists of 80 multi-turn questions, where responses are rated by GPT-4 on a 1-10 scale, with the average rating reported.

**Results.** The model performance is summarized in Table 1. Our results show that ONPO consistently outperforms or achieves comparable performance to the baselines across both base models. Among the three benchmarks, the length-controlled (LC) win rate in AlpacaEval 2.0 exhibits the highest 0.98 Spearman correlation with Chatbot Arena rankings [Dubois et al., 2024]. In this benchmark, ONPO outperforms the strongest baseline by a clear margin—achieving a 9.9% improvement on Llama-3-SFT and a 21.2% improvement on Mistral-It. These results align with our theoretical findings, demonstrating that ONPO benefits from an improved bound on the duality gap. We also compare ONPO with other LLMs that have significantly larger parameters, such as Llama-3-70B-it, Mixtral-8x22B-it and GPT-4-Turbo. Remarkably, our ONPO even outperforms models with at least nine times more parameters.

Table 2: Model performance on more academic benchmarks (AVG: average).

| Model | GPQA | Hellaswag | MMLU-Pro | Winogrande | TruthfulQA | GSM8K | AVG |
|---|---|---|---|---|---|---|---|
| Mistral-It | 30.1 | 83.5 | 30.4 | 74.2 | 59.7 | 49.5 | 54.6 |
| Iterative DPO | 29.6 | 83.3 | 28.0 | 75.1 | 64.0 | 45.7 | 54.3 |
| SPPO | 28.7 | 83.5 | 28.1 | 73.9 | 66.4 | 49.9 | 55.1 |
| INPO | 28.8 | 82.9 | 28.9 | 74.9 | 64.7 | 46.3 | 54.4 |
| ONPO | 30.4 | 83.7 | 29.9 | 75.1 | 65.5 | 47.8 | **55.4** |

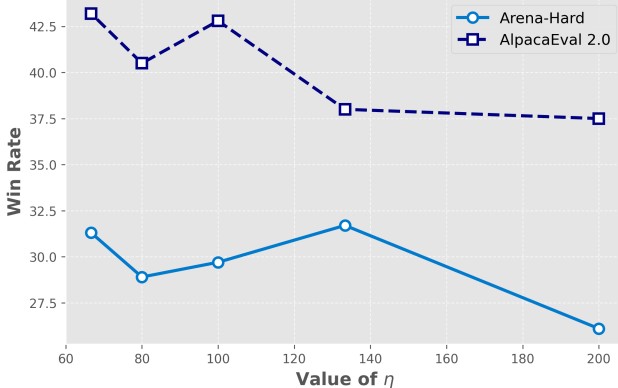

Figure 1: Performance of ONPO with different values of $\eta$ on Arena-Hard and AlpacaEval 2.0. ONPO consistently outperforms the best baseline, which achieves a win rate of 25.3 on Arena-Hard and 35.3 on AlpacaEval, respectively.

## 5.2 More Results on Academic Tasks

In this subsection, we evaluate the model's reasoning and calibration abilities across six academic benchmarks: GPQA [Rein et al., 2023] for graduate-level science question answering, MMLU-Pro [Wang et al., 2024] for multitask language understanding, Hellaswag [Zellers et al., 2019] for commonsense inference, Winogrande [Sakaguchi et al., 2021] for difficult commonsense reasoning, TruthfulQA [Lin et al., 2021] to assess the model's tendency to reproduce falsehoods, and GSM8K [Cobbe et al., 2021] for mathematical reasoning.

It is important to note that these benchmarks primarily evaluate a model's intrinsic knowledge and capabilities, which are developed during the pre-training stage rather than the alignment stage. However, as observed in prior work [Ouyang et al., 2022, OpenAI, 2023], alignment can sometimes have a negative impact on these abilities—a phenomenon known as the "alignment tax". Therefore, our purpose in presenting these results is to verify that our alignment method preserves the model's abilities rather than demonstrating performance improvements.

We show the results using Mistral-Instruct-v0.3 as the base model and compare ONPO with three baselines as well as the base model itself. The results in Table 2 show that ONPO achieves a slightly higher average performance than both the base model and the baselines, demonstrating that ONPO does not over-align the model and effectively preserves its intrinsic knowledge and abilities.

## 5.3 Hyperparameter Sensitivity Analysis

In this subsection, we analyze the sensitivity of ONPO to the hyperparameter $\eta$, which serves as the learning rate in the update rule. We conduct experiments using Mistral-Instruct-v0.3 as the base model and vary $\eta$ from $200/3$ to $200$. The results, presented in Figure 1, indicate that ONPO consistently achieves strong performance across different values of $\eta$ and outperforms the baselines, demonstrating its robustness to hyperparamter variations.

---

[5]https://huggingface.co/datasets/RLHFlow/prompt-collection-v0.1

# 6 Conclusion and Future Work

We propose Optimistic Nash Policy Optimization (ONPO), a novel approach for aligning LLMs with general preferences through self-play. By integrating optimistic online mirror descent, ONPO achieves improved theoretical guarantees for approximating the Nash policy of the game. More importantly, our experimental results demonstrate that ONPO consistently outperforms or matches state-of-the-art general preference alignment methods across multiple benchmarks. A potential limitation of this work is that our experiments focuses on the single-turn setting. In the future, we plan to explore the implementation of ONPO under the multi-turn setting.

## Acknowledgements

Nan Jiang acknowledges funding support from NSF CNS-2112471, NSF CAREER IIS-2141781, Google Scholar Award, and Sloan Fellowship.

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

## A  Related Literature

**Reward-Based RLHF.**   Since the first RLHF framework proposed by Christiano et al. [2017], RLHF has achieved tremendous success in aligning large language models (LLMs), powering models such as Instruct-GPT [Ouyang et al., 2022], Llama 2 [Touvron et al., 2023], and Claude [Bai et al., 2022b]. The RLHF pipeline typically involves training a reward model followed by applying policy gradient methods such as PPO [Schulman et al., 2017] to optimize a KL-regularized objective [Korbak et al., 2022, Li et al., 2023b]. Nevertheless, the use of PPO in RLHF introduces challenges, including instability during training [Choshen et al., 2019] and high computational costs [Yuan et al., 2023]. To address these limitations, Rafailov et al. [2024b] proposed the DPO algorithm, which directly optimizes preferences by minimizing a loss objective on offline datasets. Additionally, other direct preference learning algorithms have been developed, including offline methods [Ethayarajh et al., 2024] and online (iterative) methods [Xie et al., 2024, Xiong et al., 2024, Yuan et al., 2024]. However, all these algorithms are reward-based and rely on the Bradley-Terry (BT) model assumption. In this paper, we remove the BT model assumption and consider general preference alignment.

**RLHF with General Preferences.**   Azar et al. [2024] is the first to consider the general preference without BT model assumption. They propose the offline IPO algorithm to learn the optimal policy when the comparator policy is fixed. Munos et al. [2023] formulate the alignment problem as a two-player zero-sum game and propose the iterative Nash-MD algorithm to find the Nash policy of the game. Subsequently, there has been a line of work [Ye et al., 2024, Calandriello et al., 2024, Rosset et al., 2024, Wu et al., 2024] developing online algorithms for learning the Nash policy. The closest work related to ours is Zhang et al. [2024], which also employs a no-regret learning algorithm for self-play. However, our algorithm incorporates an optimistic predictor into the policy update, achieving improved theoretical guarantees and better empirical performance. A detailed comparison between our algorithm and other general preference alignment algorithms is provided in Section 4.

**Learning in Games.**   Online learning and self-play algorithms are widely used in approximating the equilibrium of games, including normal-form games [Freund and Schapire, 1999, Daskalakis et al., 2011, Mai et al., 2018, Roy et al., 2019, Chen and Peng, 2020, Wei et al., 2020, Daskalakis et al., 2021], extensive-form games [Zinkevich et al., 2007, Kroer et al., 2020, Kozuno et al., 2021, Lee et al., 2021, Bai et al., 2022a] and Markov games [Wei et al., 2017, Jin et al., 2021, Liu et al., 2021, Mao and Başar, 2023]. Our work is inspired by the faster convergence properties of optimistic online mirror descent in equilibrium learning [Rakhlin and Sridharan, 2013, Syrgkanis et al., 2015].

## B  Extension to the Multi-Turn Setting

In this section, we describe how ONPO can be extended to the multi-turn setting, which is formulated as a contextual Markov decision process (CMDP) [Shani et al., 2024]. The interaction between the LLM and the environment unfolds as follows: the LLM starts at a fixed initial state $s_1 \in \mathcal{S}$ and takes an action $y_1 \sim \pi(\cdot \mid s_1)$. The environment then transitions to the next state $s_2 \sim P(\cdot \mid s_1, y_1)$ according to the transition dynamics $P$, and the LLM subsequently takes action $y_2 \sim \pi(\cdot \mid s_2)$. This process repeats for $H$ steps, ultimately reaching the final state $s_{H+1}$. At the end of the interaction, the preference oracle compares two final states and provides a preference signal: $z \sim \mathrm{Ber}\big(\mathbb{P}(s_{H+1}^1 \succ s_{H+1}^2)\big)$. This CMDP formulation effectively captures various LLM applications, including chatbot interactions and token-level MDPs [Rafailov et al., 2024a].

In the multi-turn setting, the challenge is that preferences are only provided for the final states, and there is no direct feedback for intermediate states. To address this, we use Q-value functions, which capture the long-term expected outcomes, in the optimization objective. For each state $s_h$, the update rule for $\pi_{t+1}(\cdot \mid s_h)$ is:

$$\operatorname*{argmax}_{\pi} \langle \pi, Q^{\pi_t, \pi_t}(s_h, \cdot) \rangle - \frac{1}{\eta}\mathrm{KL}(\pi(\cdot \mid s_h)\|\pi_t(\cdot \mid s_h)),$$

where $Q^{\pi_t, \pi_t}(s_h, y_h) = \mathbb{E}_{\pi_t}\left[\mathcal{P}(s_{H+1} \succ \pi_t) \mid s_h, y_h\right]$ and $\mathcal{P}(s \succ \pi_t)$ represents $\mathbb{E}_{\pi_t}\left[\mathbb{P}(s \succ s_{H+1})\right]$. Here $\langle \pi, Q^{\pi_t, \pi_t}(s_h, \cdot) \rangle$ measures the probability of $\pi$ outperforming $\pi_t$ at state $s_h$. The update rule for $\pi_{t+1}'$ is similar, except that the KL divergence is computed between $\pi$ and $\pi_t'$.

The primary challenge in implementing ONPO in the multi-turn setting lies in the efficient estimation of $Q^{\pi_t, \pi_t}$. Shani et al. [2024] propose to use an actor-critic framework that employs policy-gradient

methods such as PPO [Schulman et al., 2017] for policy optimization. However, policy-gradient methods are known to exhibit high variance and sensitivity to implementation details, leading to increased computational costs. In this paper, we focus on implementing ONPO in the single-turn setting and leave the implementation under the multi-turn setting for future work.

## C  Proofs for Section 3

### C.1  Proof for Theorem 1

*Proof.* According to the regret analysis of OMD [Lattimore and Szepesvári, 2020], for any policy $\pi$, we have

$$\sum_{t=1}^{T} \langle \pi, r_t \rangle - \sum_{t=1}^{T} \langle \pi_t, r_t \rangle \leq \frac{\mathrm{KL}(\pi \| \pi_1)}{\eta} + \eta \sum_{t=1}^{T} \| r_t \|_{\infty}^2$$

$$\leq 2\sqrt{TD}.$$

The rest proof follows from Theorem 3 in Zhang et al. [2024]. □

### C.2  Proof for Theorem 2

*Proof.* Let $\psi(\pi) = \sum_y \pi(y) \log \pi(y)$, the KL divergence between $\pi_1$ and $\pi_2$ can also be written as the Bregman divergence term:

$$\mathrm{KL}(\pi_1 \| \pi_2) = D_\psi(\pi_1, \pi_2) = \psi(\pi_1) - \psi(\pi_2) - \langle \nabla \psi(\pi_2), \pi_1 - \pi_2 \rangle.$$

Since $\psi$ is strongly convex with respect to $L_1$ norm, we can apply regret analysis from Rakhlin and Sridharan [2013], Syrgkanis et al. [2015] and obtain that for any $\pi'$

$$\sum_{t=1}^{T} \langle \pi' - \pi_t, r_t \rangle \leq \frac{\mathrm{KL}(\pi' \| \pi_1')}{\eta} + \eta \sum_{t=1}^{T} \| r_t - r_{t-1} \|_{\infty}^2 - \frac{1}{4\eta} \sum_{t=2}^{T} \| \pi_t - \pi_{t-1} \|_1^2.$$

We observe that for any $t \geq 2$ and any $y$,

$$|r_t(y) - r_{t-1}(y)| = |\sum_{y'} \mathbb{P}(y \succ y')(\pi_t(y) - \pi_{t-1}(y))| \leq \| \pi_t - \pi_{t-1} \|_1.$$

Once we have $\frac{1}{4\eta} \geq \eta$, the terms $\eta \| r_t - r_{t-1} \|_{\infty}^2$ and $-\frac{1}{4\eta} \| \pi_t - \pi_{t-1} \|_1^2$ cancel out and we get

$$\sum_{t=1}^{T} \langle \pi' - \pi_t, r_t \rangle \leq 2\sqrt{D}.$$

Next, we decompose the duality gap as:

$$\mathrm{DualGap}(\bar{\pi}) = \underbrace{\max_{\pi_1} J(\pi_1, \bar{\pi}) - \frac{1}{2}}_{\text{Term A}} + \underbrace{\frac{1}{2} - \min_{\pi_2} J(\bar{\pi}, \pi_2)}_{\text{Term B}}.$$

We show how to bound Term A and Term B is bounded similarly due to the symmetric nature of the game. Let $\pi' = \mathrm{argmax}_{\pi_1} J(\pi_1, \bar{\pi})$, we have

$$J(\pi', \bar{\pi}) - \frac{1}{2} = \frac{1}{T} \sum_{t=1}^{T} J(\pi', \pi_t) - J(\pi_t, \pi_t)$$

$$= \frac{1}{T} \sum_{t=1}^{T} \langle \pi' - \pi_t, r_t \rangle$$

$$\leq \frac{2\sqrt{D}}{T}.$$

The proof is finished by also having $\frac{1}{2} - \min_{\pi_2} J(\bar{\pi}, \pi_2) \leq \frac{2\sqrt{D}}{T}$. □

# D   Additional Experiment Details

For the implementation of ONPO, we follow the hyperparameters in Dong et al. [2024], including the cosine learning rate scheduler with a peak learning rate of $5 \times 10^{-7}$, a 0.03 warm-up ratio, and a global batch size of 128. We use a grid search for $1/\eta$ over $[0.1, 0.05, 0.02, 0.01, 0.005]$ and set $1/\eta = 0.01$. Llama-3-SFT is trained for 5 iterations[6], where in each iteration $\pi'_t$ is trained for 2 epochs and $\pi_t$ for 1 epoch. While Mistral-Instruct, having already undergone instruction fine-tuning, is thereby trained for 3 iterations, with $\pi'_t$ trained for 1 epoch and $\pi_t$ for 2 epochs in each iteration. To ensure a fair comparison, all baselines are trained using the same number of iterations and the same prompt set as ONPO. All experiments are conducted on 8×A100 GPUs with 40GB memory each.

---

[6]Iteration 1, Iteration 2, Iteration 3, Iteration 4, Iteration 5.

