# OpenReview forum: "Improving LLM General Preference Alignment via Optimistic Online Mirror Descent"
_NeurIPS.cc/2025/Conference — NeurIPS 2025 spotlight_

### Official Review · Reviewer_iMtJ · 2025-06-26

**Clarity:** 3
**Significance:** 2
**Originality:** 3
**Rating:** 4
**Confidence:** 4

**Summary:**

This paper formulate LLM alignment as a two-player zero-sum game and elimate the assumption of Bradley-Terry model. The authors propose Optimistic Nash Policy Optimization (ONPO), which integrates optimistic online mirror descent into self-play to learn Nash policies. ONPO achieves improved theoretical guarantees with O(T^-1) duality gap convergence and demonstrates superior performance over state-of-the-art methods across multiple benchmarks.

**Questions:**

- Is it possible to implement the algorithm at the minibatch level? This could potentially lead to larger T, even if it introduces increased noise.

**Ethical Concerns:**

["NO or VERY MINOR ethics concerns only"]

**Final Justification:**

I appreciate the authors’ response. Their discussion of related work has addressed my concerns. I will maintain my positive score.

**Limitations:**

Yes

**Quality:**

3

**Strengths And Weaknesses:**

Strength：
- The proposed method theoretically achieves O(1/T) convergence rate with solid theoretical guarantees.
- The paper provides detailed comparison with similar methods, clearly articulating the differences and justifying the algorithm design.
- Main experiments demonstrate the algorithm's effectiveness with notable improvements over baselines.

Weaknesses：
- The proposed method requires maintaining and training two models simultaneously, introducing additional computational overhead.
- While the algorithm claims faster convergence, given the small T=3~5 in practice, it's unclear whether the theoretical analysis provides meaningful guidance for real applications (since these are just bounds).
- The paper may need to supplement comparisons under equal computational budgets, as the algorithm's T=3~ 5 might correspond to baseline methods' T=6~8, suggesting that despite theoretically faster convergence, it may not be superior under limited computational resources.
- "Results where the baseline outperforms ONPO are underlined." This method of presenting results in Table 1 can easily lead to misunderstandings.

---

> ### Author Rebuttal · Authors · 2025-07-31
>
> We thank the reviewer for their valuable comments and respond to the weaknesses and questions below.
>
> **While the algorithm claims faster convergence, given the small T=3~5 in practice, it's unclear whether the theoretical analysis provides meaningful guidance for real applications (since these are just bounds).**
>
> We would like to clarify that our theoretical bounds for both OMD and optimistic OMD share the same constant, and the improvement remains notable even for small values of $T$. For instance, at $T = 5$, the convergence rate improves from $1/\sqrt{5}$ to $1/5$, yielding a 2× gain. While this improvement becomes more pronounced as $T$ increases, it is common in the RLHF and LLM alignment literature [Xie et al., 2024; Wu et al., 2024; Zhang et al., 2024] to develop theoretical analyses for relatively large $T$, even though practical experiments often use small values such as 3 or 5.
>
>
> **The paper may need to supplement comparisons under equal computational budgets, as the algorithm's T=3~ 5 might correspond to baseline methods' T=6~8, suggesting that despite theoretically faster convergence, it may not be superior under limited computational resources.**
>
> We would like to clarify our online training setup, which follows from prior LLM alignment literature. In each iteration $t$, we construct the preference dataset $D_t$ by sampling a new set of prompts and generating responses using the current policy. Crucially, the prompt set changes across iterations. To ensure a fair comparison, we run all methods for the same number of iterations, ensuring that they are trained on the same set of prompts.
>
> Regarding computational cost, all baselines are trained for two epochs per iteration. In the Mistral experiments, ONPO uses the same total number of training epochs, one for $\pi'\_{t+1}$ and one for $\pi_{t+1}$, resulting in the same computational cost. In the LLaMA experiments, we use two epochs for $\pi'\_{t+1}$ and one for $\pi_{t+1}$, adding just one extra epoch per iteration. However, the majority of computation time per iteration (approximately 3–4 hours) is spent on generating the preference dataset, and the additional epoch only adds about 1.5 hours, which is relatively minor. Moreover, we experimented with training the baselines for three epochs per iteration but did not observe performance improvement.
>
> **The proposed method requires maintaining and training two models simultaneously, introducing additional computational overhead.**
>
> We would like to clarify that $\pi'\_{t+1}$ and $\pi_{t+1}$ are trained sequentially rather than simultaneously. After constructing the preference dataset $D_t$, we first minimize $g_t$ to obtain $\pi'\_{t+1}$, and then minimize $g_{t+1}$ to obtain $\pi_{t+1}$. The log ratio of $\pi'\_{t+1}$ in the objective $g_{t+1}$ is fixed and precomputed. As a result, there is no need to maintain both models in memory at the same time, and the training does not introduce additional memory overhead from simultaneously handling two models.
>
>
> **Results where the baseline outperforms ONPO are underlined." This method of presenting results in Table 1 can easily lead to misunderstandings.**
>
> Thank you for pointing this out. We agree that the current presentation could be misleading and will revise it in the next version.
>
> **Is it possible to implement the algorithm at the minibatch level? This could potentially lead to larger T, even if it introduces increased noise.**
>
> We assume the reviewer is referring to using only a minibatch of prompts in each iteration to train the model. While this approach could reduce the computational cost per iteration and allow for a larger $T$, a small set of prompts may not provide sufficient training signal to accurately obtain $\pi_{t+1}$, which serves as the target policy in mirror descent. Our current online training setup follows prior RLHF literature, which typically uses a large batch of prompts per iteration. That said, exploring a minibatch-level implementation is an interesting direction for future work.

---

> > ### Comment · Reviewer_iMtJ · 2025-08-07
> >
> > Thank you for the response. After carefully reviewing the rebuttal, I have decided to maintain my original positive score.

---

### Official Review · Reviewer_eFXw · 2025-06-27

**Clarity:** 2
**Significance:** 3
**Originality:** 3
**Rating:** 4
**Confidence:** 4

**Summary:**

The paper introduces Optimistic Nash Policy Optimization (ONPO), a novel online algorithm designed for aligning large language models with general human preferences beyond the restrictive Bradley-Terry assumption. By framing the preference alignment task as a two-player zero-sum game, ONPO leverages optimistic online mirror descent (OMD) to approximate the Nash equilibrium more efficiently. Theoretically, ONPO improves the convergence rate on the duality gap from the previously established and ensures linear convergence in the final iterate. Empirically, ONPO demonstrates superior performance compared to existing methods like iterative DPO, SPPO, and INPO across standard benchmarks including AlpacaEval 2.0, Arena-Hard, and MT-Bench, effectively maintaining model capabilities on reasoning and calibration tasks.

**Questions:**

See weaknesses. I am willing to raise my score if my concerns are addressed.

**Ethical Concerns:**

["NO or VERY MINOR ethics concerns only"]

**Final Justification:**

I thank the authors for the additional experiments and for implementing both Nash-MD and Online IPO during the rebuttal. While these results are helpful, my main concerns remain. The lack of comprehensive statistical significance reporting across all benchmarks is a notable limitation, especially for RL-based methods where variance can be high. Additionally, although the new baselines were included, the comparisons are still somewhat limited in scope, and the absence of broader variance analysis makes it difficult to fully assess the reliability of the reported improvements. I therefore keep my original rating.

**Limitations:**

yes

**Paper Formatting Concerns:**

-

**Quality:**

2

**Strengths And Weaknesses:**

### Strengths

* **Theoretical Advances**: The paper proposes Optimistic Nash Policy Optimization (ONPO), an innovative integration of optimistic online mirror descent (OMD) into LLM alignment. This method notably improves the convergence rate on the duality gap from $O(T^{-1/2})$ to $O(T^{-1})$, a significant theoretical advance.

* **Strong Empirical Results**: Empirical evaluations demonstrate consistent improvements over state-of-the-art alignment methods (Iterative DPO, SPPO, INPO) across prominent benchmarks (AlpacaEval 2.0, Arena-Hard, MT-Bench). Notably, ONPO achieves substantial improvements on AlpacaEval 2.0, outperforming baseline methods even when compared to significantly larger models.

### Weaknesses

* **Lack of Statistical Significance Reporting**: The paper presents experimental results without reporting standard deviations or confidence intervals, casting doubt on the statistical significance of the observed improvements. Without this information, it is challenging to determine whether differences between methods are statistically meaningful or simply the result of random variance.

* **Missing Empirical Comparison with Closely Related Methods**: Although the discussion section thoroughly situates ONPO among related algorithms like Nash-MD, DNO, and Online IPO, these methods are not included in the empirical comparison. This omission weakens the paper’s claim of superiority, especially since these algorithms are based on similar game-theoretic or general preference formulations.

---

> ### Author Rebuttal · Authors · 2025-07-31
>
> We thank the reviewer for their valuable comments and respond to the weaknesses and questions below.
>
> **Lack of Statistical Significance Reporting**
>
> We acknowledge the importance of statistical significance as raised by the reviewer. However, in contrast to traditional machine learning, LLM experiments are extremely resource-intensive, making statistical significance reporting relatively rare in the LLM community. This is not only the case in academic work, including closely related papers such as Iterative DPO [Dong et al., 2024], SPPO [Wu et al., 2024], and INPO [Zhang et al., 2024], but also in large-scale technical reports such as the Qwen3 Technical Report [Yang et al., 2025].
> Here we report standard deviations on the AlpacaEval 2.0 benchmark, one of the most widely used and representative alignment benchmarks:
>
> | Method                  | AlpacaEval 2.0 (LLaMA-3-SFT) | AlpacaEval 2.0 (Mistral-IT) |
> |-------------------------|------------------------------|------------------------------|
> | Iterative DPO           | 28.3 ± 1.3                   | 32.0 ± 1.3                   |
> | SPPO                    | 38.5 ± 1.4                   | 33.1 ± 1.2                   |
> | INPO                    | 44.2 ± 1.4                   | 35.3 ± 1.4                   |
> | ONPO                    | **48.6 ± 1.5**               | **42.8 ± 1.4**               |
>
>
> As shown, the standard deviations are relatively small and ONPO consistently outperforms all baselines by a statistically meaningful margin.
>
> **Missing Empirical Comparison with Closely Related Methods**
>
> We address the reviewer’s concern by discussing each method individually:
>
> **Nash-MD**. The Nash-MD algorithm cannot be directly implemented in practice for two main reasons: (1) computing the geometric mixture policy is computationally intractable, and (2) estimating the expected win rate between two policies is also challenging in practice. As a result, practical implementations of Nash-MD require heuristic approximations. At the time of our project, we did not find the official implementation, and prior related works did not include comparisons with Nash-MD.
>
> During the rebuttal period, we made our best effort to implement a variant of Nash-MD using the TRL trainer. Since our setting provides only binary preference signals rather than scalar win probabilities, we approximated the win rate by assigning a value of 1 to the preferred response and 0 to the other. The results on LLaMA3-8B-SFT are shown below.
> | Model               | AlpacaEval 2.0 | Arena-Hard | MT-Bench |
> |---------------------|----------------|------------|----------|
> | Nash-MD + LLaMA-3-SFT | 34.0           | 26.8       | 8.16     |
> | ONPO + LLaMA-3-SFT  | 48.6           | 36.4       | 8.40     |
>
> As shown in the table, ONPO consistently outperforms Nash-MD across all three evaluation benchmarks.
>
> **DNO**. As stated in lines 224–232, the original DNO algorithm is not directly implementable in practice. Rosset et al. [2024] proposed a practical variant, DNO-Prct, which replaces the intractable update with the DPO objective. As a result, DNO-Prct can be interpreted as an iterative version of DPO and our experiments already include a comparison with Iterative DPO.
>
> **Online IPO**. There are two possible ways to implement online IPO. One is to apply the same loss transformation technique used in IPO and INPO; in this case, online IPO reduces to INPO without the KL stability term, which has been shown to underperform compared to INPO, see Table 3 in Zhang et al. [2024]. The other option is to use policy gradient methods to directly optimize the objective. However, as discussed in lines 215–223, policy gradient methods often suffer from high variance and can lead to unstable learning. Furthermore, we could not find any official policy gradient implementation of online IPO. We believe this is also why prior works have not included it in empirical comparisons.
>
> Taken together, we would like to emphasize that we have already compared ONPO with state-of-the-art preference alignment methods, including SPPO and INPO, demonstrating the effectiveness of our approach.

---

> > ### Comment · Reviewer_eFXw · 2025-08-05
> >
> > Thank you to the authors for the clarifications and the additional experiments.
> >
> > 1) While it is true that some papers in the field do not perform formal statistical testing, I still view the lack of statistical significance analysis as a major limitation of all such works, regardless of the authors' reputation. Most papers do report standard deviations alongside performance metrics—especially in the context of developing new RL-based algorithms, where instability and variance are common challenges.
> >
> > 2) You are correct that a version of Nash-MD is available in TRL, and I appreciate the inclusion of new evaluation results. However, the lack of an official implementation of Online IPO should not, in itself, justify excluding it as a baseline, especially given that adapting Online DPO into Online IPO is relatively straightforward by modifying the loss function.
> >
> > I will maintain my original score. Thank you again to the authors for the rebuttal.

---

> > > ### Author Response · Authors · 2025-08-05
> > >
> > > Thank you for your response. We would like to further clarify the point regarding Online IPO. If by implementing it you are referring to adapting Online DPO by replacing the loss with the IPO objective, this corresponds exactly to the INPO algorithm without the KL stability term, as noted in our rebuttal. According to Table 3 in Zhang et al. [2024], this variant is outperformed by INPO. Additionally, SPPO has been compared against iterative IPO and shown to outperform it as well (see Table 1 in Wu et al. [2024]).
> > >
> > > Since our method consistently outperforms both INPO and SPPO, we believe this provides strong evidence that ONPO is also superior to Online IPO, especially given the time constraints during the rebuttal period.

---

> > > > ### Author Response · Authors · 2025-08-09
> > > >
> > > > We implemented Online IPO in the manner suggested by the reviewer.
> > > > | Model               | AlpacaEval 2.0 | Arena-Hard | MT-Bench |
> > > > |---------------------|----------------|------------|----------|
> > > > | Online IPO + LLaMA-3-SFT | 38.9           | 34.7       | 8.21     |
> > > > | ONPO + LLaMA-3-SFT  | 48.6           | 36.4       | 8.40     |
> > > >
> > > > As shown in the table, ONPO consistently outperforms Online IPO across all three benchmarks, with a particularly large margin on AlpacaEval 2.0. The standard deviations for the two methods on this benchmark are 1.4 and 1.5, respectively, indicating that the performance difference is statistically significant.

---

### Official Review · Reviewer_9tRL · 2025-07-02

**Clarity:** 4
**Significance:** 4
**Originality:** 4
**Rating:** 5
**Confidence:** 3

**Summary:**

This paper addresses the problem of aligning large language models (LLMs) with human preferences without relying on the restrictive Bradley-Terry (BT) model assumption. The authors formulate LLM alignment as a two-player zero-sum game and propose Optimistic Nash Policy Optimization (ONPO), which integrates optimistic online mirror descent into a self-play framework. The authors obtain an improved theoretical convergence guarantees of $\mathcal{O}\left(\frac{1}{T}\right)$ duality gap bound versus previous $\mathcal{O}\left(\frac{1}{\sqrt{T}}\right)$ results. The authors also prove a linear convergence in the last iterate which is a novel result for this setup. Finally the authors provide strong empirical performance across multiple benchmarks including AlpacaEval 2.0, Arena-Hard, and MT-Bench when the other models have same/comparable number of parameters.

**Questions:**

How much more computational resources would be needed for a multi turn setup?

**Ethical Concerns:**

["NO or VERY MINOR ethics concerns only"]

**Final Justification:**

I have gone over the replies of the reviewers. They have answered the questions to my satisfaction, and I maintain my positive score.

**Limitations:**

The assumption of an oracle and single turn setup seem to be the only limitations I can think of. However, for a proof of concept paper these seem to be reasonable assumptions.

**Quality:**

4

**Strengths And Weaknesses:**

Strengths:
1.The paper identifies a key limitation of existing RLHF methods that rely on BT assumptions and provides evidence from prior literature that challenges its validity in modeling human preferences.
2. Achieves state of the art convergence results for the duality gap as well as linear convergence rate for the last iterate. The latter is something that is novel.
3. Dropping the Bradley-Terry model assumption is well-justified. The authors provide compelling evidence that human preferences often violate transitivity assumptions, especially at population levels, making the general preference setting more realistic.

Weakness:
1. While performance improvements against other models have been established, they have been done so for models of size 8B/7B. the question as to whether the same performance improvement would be had if the model sizes are scaled up to larger sizes remains open.
2. Their is an assumption of a preference oracle here. Is that a realistic assumption? It appears not to be the case but I am open to the authors explaining why it might be.
3. The self play here is done for a single turn. While I am not very familiar with the practical aspects of RLHF this seems significant.

---

> ### Author Rebuttal · Authors · 2025-07-31
>
> We thank the reviewer for their valuable comments and respond to the weaknesses and questions below.
>
> **While performance improvements against other models have been established, they have been done so for models of size 8B/7B. the question as to whether the same performance improvement would be had if the model sizes are scaled up to larger sizes remains open.**
>
> Due to limited computational resources, our experiments are conducted on 7B/8B base models. This limitation is common in the LLM alignment literature, where experiments on larger-scale models are rarely presented. We acknowledge that it remains an open question whether the same performance gains hold at larger model scales, and we plan to explore this direction in future work as resources allow.
>
> **Their is an assumption of a preference oracle here. Is that a realistic assumption? It appears not to be the case but I am open to the authors explaining why it might be.**
>
> We would like to clarify that the preference oracle is not an assumption but a general formulation of the underlying human preferences we aim to align with. Specifically, given a prompt $x$ and two responses $y_1$ and $y_2$, the oracle gives the probability $\mathbb{P}(y_1 \succ y_2 \mid x)$ that humans would prefer $y_1$ over $y_2$. This offers a  mathematical abstraction of the process that humans generate preference labels.
>
> Commonly used models such as the Bradley–Terry (BT) model is a restrictive special case of the general preference formulation, where $\mathbb{P}(y_1 \succ y_2 \mid x)$ is assumed to follow a specific parametric form, e.g., the sigmoid of the reward difference $R(x, y_1) - R(x, y_2)$. In contrast, we only assume access to a binary preference signal sampled from $\mathrm{Ber}(\mathbb{P}(y_1 \succ y_2 \mid x))$.  As long as we are comfortable with modeling the human feedback as a probability distribution, this is the **minimal** assumption. This binary signal plays a similar role to labels in supervised learning, it is essential for model training, and all LLM alignment methods rely on such preference data for supervision.
>
> **The self play here is done for a single turn. While I am not very familiar with the practical aspects of RLHF this seems significant.**
>
> As discussed in Appendix B, our method can be extended to the multi-turn setting. However, implementing the multi-turn setting would necessitate using deep reinforcement learning techniques and policy gradient methods, which are known to be computationally expensive and unstable during training. Our primary goal in this work is to develop a computationally lightweight method and focus on the single-turn setting, leaving the implementation of the multi-turn extension as future work.
>
> **How much more computational resources would be needed for a multi turn setup?**
>
> We do not have precise estimates for the additional computational resources required for a multi-turn setup. However, based on our experience from related projects involving policy gradient methods for post-training large language models, achieving satisfactory performance typically requires 10 or more epochs. In contrast, our current method uses only 2–3 epochs per iteration.

---

### Official Review · Reviewer_9enM · 2025-07-03

**Clarity:** 2
**Significance:** 2
**Originality:** 3
**Rating:** 4
**Confidence:** 4

**Summary:**

The paper proposes a method for preference alignment of LLMs by formulating the problem as a two-player game via game theory. This integrates online mirror descent into an alignment framework to approximate the Nash policy. The goal is to leverage the faster convergence properties of online mirror descent in the preference learning use case.

**Questions:**

- Even without full implementation, demonstrating ONPO on multi-turn with toy examples would help validate its broader application and scope.
- Runtime and efficiency comparisons. Training time, GPU setup, convergence speed, against more relevant baselines listed above. This will help validate the claim of "computational lightness".
- Add discussion/experiments considering the possibility of a noisy oracle, to try and model how this will behave in real world.

**Ethical Concerns:**

["NO or VERY MINOR ethics concerns only"]

**Final Justification:**

The rebuttal clarified certain points, yet detracted from others. The authors concede experiments are on "toy examples" which is concerning. The main practical deployment possibilities of this method are not apparent, especially in this particularly dense area of research where many topics have been published in recent months. I would like to state my overall score as **3.5** ideally. Although the authors do not demonstrate a practical application where their work excels at, the concept of incorporating traditional mirror descent is interesting. However this work will significantly benefit from an additional round of reviews as well, to clearly solidify their theoretical contributions/practical superiority in a specific use-case.

**Limitations:**

yes

**Quality:**

2

**Strengths And Weaknesses:**

Strengths:
- Paper aims to integrate classical mirror descent with preference learning. This is an interesting and novel approach.
- Optimistic Nash Policy Optimization improving the convergence rate on the duality gap with additional guarantees is interesting and non-trivial. The theoretical contribution does integrate well with the paper, and shows understanding of learning with general preferences beyond the Bradley-Terry framework.
- Compares against DPO

Weaknesses:
- Only compares against DPO and 2 minor methods. Other methods in this area (SimPO, ORPO, R-DPO, RL-based methods etc) are not explored.
- No discussion of experimentation setup. Hardware, training time, codebase, etc. Makes it hard to assess practical trade-offs between methods.
- Method assumes access to a general preference oracle (possibly learned), but does not consider the effect of noisy/imperfect oracle feedback. Since human preferences can be inconsistent, this is a critical.

---

> ### Author Rebuttal · Authors · 2025-07-31
>
> We thank the reviewer for their valuable comments and respond to the weaknesses and questions below.
>
> **Only compares against DPO and 2 minor methods. Other methods in this area (SimPO, ORPO, R-DPO, RL-based methods etc) are not explored.**
>
> We would like to clarify that the two baselines we compare against, SPPO and INPO, are state-of-the-art approaches for online preference alignment.
>
> The methods mentioned by the reviewer are designed for the **offline** alignment setting, where the algorithm optimizes a learning objective over a fixed, pre-collected dataset. In contrast, our work focuses on the **online** setting, where no response pairs are pre-collected and the model must iteratively generate responses and improve itself based on collected preferences.
> As stated on line 263, prior work has shown that online or iterative alignment methods can outperform their offline counterparts [Guo et al., 2024; Dong et al., 2024]. For this reason, we compare ONPO with other online methods to ensure a fair and meaningful evaluation.
>
> Among the offline methods, SimPO has demonstrated strong performance. To further strengthen our comparison, we implemented an iterative variant of SimPO, where the current model generates responses and optimizes the SimPO objective at each iteration. As shown below, ONPO significantly outperforms it across all three evaluation benchmarks:
> | Model               | AlpacaEval 2.0 | Arena-Hard | MT-Bench |
> |---------------------|----------------|------------|----------|
> | SimPO + LLaMA-3-SFT | 35.0           | 31.7       | 8.19     |
> | ONPO + LLaMA-3-SFT  | 48.6           | 36.4       | 8.40     |
>
>
> **No discussion of experimentation setup. Hardware, training time, codebase, etc. Makes it hard to assess practical trade-offs between methods.**
>
> Details of our experimental setup are provided in Section 5.1 and Appendix D. As noted on line 564, all experiments were conducted on 8×A100 GPUs with 40GB memory each.
>
> We apologize for not reporting the actual training time in the main text. All methods follow the same online training setup and differ only in their learning objectives. In each iteration, approximately 3–4 hours are spent generating the preference dataset. Each baseline is then trained for two epochs, which takes around 3 hours. In the Mistral experiments, ONPO uses the same total number of training epochs per iteration, one for $\pi'\_{t+1}$ and one for $\pi_{t+1}$, resulting in equivalent computational cost. In the LLaMA experiments, we use two epochs for $\pi'\_{t+1}$ and one for $\pi_{t+1}$, introducing only one additional epoch (~1.5 hours). We also tested training the baselines with three epochs per iteration and did not observe performance improvements.
>
> Our codebase is adapted from Dong et al. [2024], and all baselines are implemented following their official code.
>
> **Method assumes access to a general preference oracle (possibly learned), but does not consider the effect of noisy/imperfect oracle feedback. Since human preferences can be inconsistent, this is a critical.**
>
> We want to first emphasize that our preference oracle is a general formulation of the underlying human preferences we aim to align with. It is not “learned” as the reviewer suggested, since the oracle is simply a mathematical abstraction of the process that human generates preference labels. Specifically, given a prompt $x$ and two responses $y_1$ and $y_2$, the oracle gives the probability $\mathbb{P}(y_1 \succ y_2 \mid x)$ that humans would prefer $y_1$ over $y_2$.
>
> It is unclear to us what the reviewer means by “inconsistent”. The reviewer might be thinking of properties of commonly used models such as the Bradley-Terry (BT) model, which assumes that $\mathbb{P}(y_1 \succ y_2 \mid x)$ follows a sigmoid function of some underlying reward/utility function $R$. In this case, a real human feedback may indeed be “inconsistent” with the BT model in that the feedback may not be consistent with any reward function.  In contrast, we only assume access to a binary preference signal sampled from $\mathrm{Ber}(\mathbb{P}(y_1 \succ y_2 \mid x))$.  As long as we are comfortable with modeling the human feedback as a probability distribution, this is really the **minimal** assumption. This binary signal plays a similar role to labels in supervised learning, it is essential for model training, and all LLM alignment methods rely on such preference data for supervision.
>
> We acknowledge that modeling and mitigating the effect of noise in preference signals is an important direction for future work. In our case, stochastic noise will simply change the function $\mathbb{P}(y_1 \succ y_2 \mid x)$, and our algorithm will still align with the preference function modified by noise. How to align with the “original” preference is analogous to handling noisy or corrupted labels in supervised learning. However, we consider this direction to be orthogonal to our primary focus, which is to develop a more effective alignment method with provably faster convergence. We will include a discussion of this point in the revised version.
>
> **Even without full implementation, demonstrating ONPO on multi-turn with toy examples would help validate its broader application and scope.**
>
> We appreciate the suggestion. However, the core challenge of the LLM alignment problem lies in enabling large models to learn complex human preferences, which are difficult to meaningfully represent in toy examples. As discussed in Appendix B, extending our method to the multi-turn setting would require incorporating deep reinforcement learning and policy gradient techniques, which are computationally expensive and often unstable. Given our focus on developing a computationally lightweight method, we chose to concentrate on the single-turn setting in this work and leave the implementation and evaluation of the multi-turn extension to future work.
>
> Shangmin Guo, Biao Zhang, Tianlin Liu, Tianqi Liu, Misha Khalman, Felipe Llinares, Alexandre Rame, Thomas Mesnard, Yao Zhao, Bilal Piot, et al. Direct language model alignment from online ai feedback. arXiv preprint arXiv:2402.04792, 2024.
>
> Hanze Dong, Wei Xiong, Bo Pang, Haoxiang Wang, Han Zhao, Yingbo Zhou, Nan Jiang, Doyen Sahoo, Caiming Xiong, and Tong Zhang. Rlhf workflow: From reward modeling to online rlhf. arXiv preprint arXiv:2405.07863, 2024.

---

### Note · Authors · 2025-08-14

Dear Area Chairs,

We sincerely thank you for your time and effort in handling our submission. Our initial reviews were 5 (accept), 4, 4 (borderline accept), and 3 (borderline reject). After the rebuttal, reviewer 9enM — who initially gave a 3 — expressed satisfaction with our clarifications and stated that they will improve the score. We kindly hope the AC can take this into consideration when making the final decision.

We truly appreciate your work in coordinating the review process and for your attention to these details.

---

### Decision · Program_Chairs · 2025-09-17

**Decision:**

Accept (spotlight)

**Comment:**

This work proposes a new RLHF method based on the optimism principle that outperforms several common benchmarks such as DPO. Theoretical analysis and experimental evaluation was sufficient to convince the reviewers of the merits of the proposed approach, to which I agree.